



# Glacier geometry and flow speed determine how Arctic marine-terminating glaciers respond to lubricated beds

Whyjay Zheng[1, 2]

[1]Department of Earth and Atmospheric Sciences, Cornell University, Ithaca, NY, USA
[2]Department of Statistics, University of California Berkeley, Berkeley, CA, USA

**Correspondence:** Whyjay Zheng (whyjz@berkeley.edu)

**Abstract.** Basal conditions directly control the glacier sliding rate and the dynamic discharge of ice flow. Recent glacier desta­bilization events indicate that some marine-terminating glaciers quickly respond to lubricated beds with increased flow speed, but the underlying physics, especially how this vulnerability relates to glacier geometry and flow characteristics, remains un­clear. This paper presents a 1-D physical framework for glacier dynamic vulnerability assuming sudden basal lubrication as an

initial perturbation. In this new model, two quantities determine the scale and the areal extent of the subsequent thinning and acceleration after the bed is lubricated: Péclet number ($P_e$) and the product of glacier speed and thickness gradient (dubbed $J_0$ in this study). To validate the model, this paper calculates $P_e$ and $J_0$ using multi-sourced data from 1996–1998 for out­let glaciers in Greenland and Austfonna Ice Cap, Svalbard, and compares the results with the glacier speed change during 1996/1998–2018. Glaciers with lower $P_e$ and $J_0$ are more likely to accelerate during this 20-year span than those with higher

$P_e$ and $J_0$, which matches the model prediction. A combined factor of ice thickness, surface slope, and initial speed for ice flow physically determines how much and how fast glaciers respond to lubricated beds, as forms of speed, elevation, and terminus change.

## 1 Introduction

Marine-terminating glaciers around the world have undergone significant acceleration, retreat, and mass loss in past decades

(e.g. Vaughan et al., 2013; Cook et al., 2016; Carr et al., 2017; Catania et al., 2020; Williams et al., 2021). At the Greenland Ice Sheet (GrIS) where a good portion of marine-terminating glaciers are studied, dynamic discharge accounts for 66% of the ice loss, while surface mass balance (SMB) accounts for the rest of 34% (Mouginot et al., 2019). For the other Arctic regions outside of the Greenland Ice Sheet, SMB contributes more mass loss than dynamic discharge (Catania et al., 2020), but there also exists several rapid acceleration events that dominate the local land ice budget (e.g. McMillan et al., 2014; Strozzi et al.,

2017a; Willis et al., 2018; Haga et al., 2020).

The acceleration and dynamic thinning of marine-terminating glaciers have been attributed to at least two different sources: basal lubrication driven by surface melt accessing to the bed, and terminus perturbation driven by ice-ocean interactions (Carr et al., 2013). Multiple observations suggest the warming of subsurface waters as the primary and widespread driver across the outlet glaciers in GrIS (e.g. Nick et al., 2009; Walsh et al., 2012; Tedstone et al., 2013; Catania et al., 2020; Wood et al., 2021;





Williams et al., 2021). As a result, modeling of glacier dynamic loss for estimating sea-level rise usually focuses on the glacier
terminus and overlooks the changing basal conditions (e.g. Nick et al., 2013). Outside of the GrIS, the drivers of the dynamic
ice loss and their relative importance remain largely uncertain (Carr et al., 2017; Strozzi et al., 2017b). Melt-induced speedups
are still suggested to be the case for several events around the Arctic, including the GrIS (e.g. Sundal et al., 2011; Dunse et al.,
2015; Zheng et al., 2019; Seddik et al., 2019). To date, the response to basal lubrication is mostly studied at a seasonal scale
(e.g. Zwally et al., 2002; Bartholomew et al., 2010; Sundal et al., 2013; Hewitt, 2013; Rathmann et al., 2017; King et al., 2018;
Williams et al., 2020). Although some speedup events seem to permanently change the basal conditions via creating a highly
crevassed glacier surface which makes meltwater reach the bed more easily (e.g. Dunse et al., 2015; Strozzi et al., 2017a;
Zheng et al., 2019; Sánchez-Gámez et al., 2019), the interannual impact of basal lubrication is still less constrained (Kehrl
et al., 2017). In addition, glacier geometry plays a vital role in how a glacier responds to an external perturbation (Carr et al.,
2017), but only terminus disruption has been physically well documented and explained (McFadden et al., 2011; Felikson
et al., 2017, 2021). Whether some glaciers are more sensitive to basal lubrication than others due to their geometry is not well
known.

To better understand how much and how fast a glacier responds to basal lubrication and its relationship to glacier geometry,
this paper presents a physical model with a 1-D framework along flowlines formulating the subsequent change (in terms of
both glacier speed and surface elevation) after the glacier bed is suddenly lubricated. We use an existing glacier perturbation
model (Zheng et al., 2019) and replace the initial thinning condition with a step reduction of basal friction along the glacier
channel. We identify key parameters that dominate elevation change rate and ice flow acceleration in this new model. Then,
using data from the Greenland Ice Sheet and Austfonna Ice Cap, Svalbard, we derive these parameters for each glacier basin
we analyze and compare them with glacier speed change over 20 years. The entire data processing workflows, including data
fetching, deriving key parameters, and scripts for making figures, are available on Github (https://github.com/whyjz/pejzero,
Zenodo DOI: https://doi.org/10.5281/zenodo.5641953) and are complied as a Jupyter Book ready to be cloud executed using
the MyBinder service for full reproducibility (Project Jupyter et al., 2018; Executable Books Community, 2020).

## 2   Model development

We build the model on the perturbation theory developed by Nye (1963), Bindschadler (1997), Felikson et al. (2017), and
Zheng et al. (2019). Our goal in this section is to formulate the change rate of ice elevation ($\frac{dh}{dt}$) and ice speed ($\frac{dU}{dt}$) after a
glacier bed is lubricated permanently. In this new model, basal lubrication is considered as a sudden perturbation without initial
elevation change. The variables defined in the model are listed in Table 1.

### 2.1   Perturbation due to a permanent change of basal conditions

We set up a glacier with the following initial values along its 1-D flowline profile: speed ($U_0$), thickness ($H_0$), flux ($q_0$), surface
slope ($\alpha_0$), and bed friction term ($K_0$). These values vary along with the flowline distance $x$ (positive towards downstream)





**Table 1.** Variables used in perturbation model defined in this study. In the dimension column, $L$ is length and $T$ is time.

| Variable | Definition | Dimension |
|---|---|---|
| $x$ | Distance along a 1-D glacier flowline towards terminus | $L$ |
| $t$ | Time after perturbation | $T$ |
| $m$ | Flow law constant | none |
| $U_0(x)$ | Glacier speed before perturbation | $LT^{-1}$ |
| $U_1(x,t)$ | Change of glacier speed after perturbation | $LT^{-1}$ |
| $K_0(x)$ | Basal friction term before perturbation | $L^{m-1}T^{-1}$ |
| $K_1(x,t)$ | Change of Basal friction after perturbation | $L^{m-1}T^{-1}$ |
| $H_0(x)$ | Glacier thickness before perturbation | $L$ |
| $H_1(x,t)$ | Change of glacier thickness after perturbation | $L$ |
| $\alpha_0(x)$ | Glacier slope before perturbation | none |
| $\alpha_1(x,t)$ | Change of glacier slope after perturbation | none |
| $q_0(x)$ | Flux before perturbation | $L^2T^{-1}$ |
| $q_1(x,t)$ | Change of flux after perturbation | $L^2T^{-1}$ |
| $C_0(x)$ | $\equiv \partial q_0/\partial H$ | $LT^{-1}$ |
| $D_0(x)$ | $\equiv \partial q_0/\partial \alpha$ | $L^2T^{-1}$ |
| $J_0(x)$ | $\equiv C_0 H_0' + D_0 \alpha_0'$ | $LT^{-1}$ |
| $P_e(x)$ | Péclet number, see Eq. 14 | none |
| $\ell$ | Characteristic length (length of perturbation) | $L$ |

and do not vary with time (i.e., steady state). We assume that the ice is purely sliding on the bed and does not have any internal deformation, that is,

$$q_0 = U_0 H_0. \tag{1}$$

The glacier speed can be further represented using the hard-bed sliding law (Weertman, 1957):

$$U_0 = K_0 H_0^m \alpha_0^m, \tag{2}$$

where $m$ is the flow-law constant and is set to 3 in this study.

At $t = 0$, the bed condition changes and the friction term becomes $K_0 + K_1$. The second term $K_1$ denotes the amount of change and is positive for a lubrication scenario. There is no initial elevation change associated with this event. We also assume this is a one-time change and is uniform over the glacier, so $K_1(x,t)$ is a constant. The subsequent change of speed, elevation,

flux, and slope, are represented as $U_1$, $H_1$, $q_1$, and $\alpha_1$ respectively. Unlike $K_1$, these quantities vary along the glacier channel





and over time. Assuming zero local surface mass balance and zero local stress imbalance (e.g. Felikson et al., 2017; Zheng et al., 2019), the conservation of mass can be expressed as:

$$\frac{\partial H_1}{\partial t} = -\frac{\partial q_1}{\partial x}. \tag{3}$$

Taking the total derivative of $q_1$ with respect to $t$ yields

$$q_1 = \frac{\partial q_0}{\partial K}K_1 + \frac{\partial q_0}{\partial H}H_1 + \frac{\partial q_0}{\partial \alpha}\alpha_1. \tag{4}$$

If we assume a much more gentle slope of the bedrock than that of the ice surface (Felikson et al., 2017; Zheng et al., 2019), the surface slope can be also expressed as the first derivative of ice thickness:

$$\alpha_1 = -\frac{\partial H_1}{\partial x} \tag{5}$$

Plugging Eqs. 1, 2, 4, and 5 into Eq. 3 yields

$$\frac{\partial H_1}{\partial t} = -\frac{K_1}{K_0}(C_0 \frac{\partial H_0}{\partial x} + D_0 \frac{\partial \alpha_0}{\partial x}) - \frac{\partial C_0}{\partial x}H_1 - (C_0 - \frac{\partial D_0}{\partial x})\frac{\partial H_1}{\partial x} + D_0 \frac{\partial^2 H_1}{\partial x^2}, \tag{6}$$

where

$$C_0 = \frac{\partial q_0}{\partial H} \tag{7}$$

and

$$D_0 = \frac{\partial q_0}{\partial \alpha}. \tag{8}$$

Since $H_1(t=0,x)=0$,

$$\frac{\partial H_1}{\partial t}|_{t=0} = -\frac{K_1}{K_0}J_0, \tag{9}$$

where

$$J_0 = C_0 H_0' + D_0 \alpha_0' = \frac{\partial q_0}{\partial H}\frac{\partial H_0}{\partial x} + \frac{\partial q_0}{\partial \alpha}\frac{\partial \alpha_0}{\partial x}. \tag{10}$$



## 2.2 $J_0$ and Péclet number ($P_e$)

Equation 9 indicates that the ratio of $K_1$ to $K_0$ and the value of $J_0$ both determine the initial elevation change rate. In a lubricating scenario, both $K_1$ and $K_0$ are positive, and $J_0$ is inversely proportional to $\frac{dh}{dt}$.

To relate $J_0$ to the glacier speed change, we start from the change of flux and assume $U_1 >> H_1$. This can be justified by many observations of glacier destabilization since the amount of speed change is usually one to two orders of magnitude higher than the amount of elevation change (e.g. McMillan et al., 2014; Willis et al., 2018). Therefore,

$$q_1 = U_1 H_0 + U_0 H_1 \approx U_1 H_0. \tag{11}$$

Since $\frac{\partial K_1}{\partial t} = 0$, we can derive the glacier speed change using Eqs. 4, 5, 7, 8, and 11:

$$\frac{\partial U_1}{\partial t} = \frac{1}{H_0} \frac{\partial q_1}{\partial t} = \frac{1}{H_0} (C_0 \frac{\partial H_1}{\partial t} - D_0 \frac{\partial}{\partial x} \frac{\partial H_1}{\partial t}). \tag{12}$$

At $t = 0$,

$$\frac{\partial U_1}{\partial t}\big|_{t=0} = -J_0 \frac{K_1}{K_0} \frac{C_0}{H_0} \tag{13}$$

Similar to the elevation change, $J_0$ is inversely proportional to the initial glacier acceleration. If two glacier beds are lubricated with the same amount of $K_1/K_0$, the glacier with a higher amount of $|J_0|$ will be more vulnerable to basal lubrication as the initial speed and elevation change rates are higher.

Also, from Eq. 9 we can predict a subsequent elevation change after $t = 0$. At this point, the last three terms in Eq. 6 begin to take part in the elevation change rate. The second term of Eq. 6 represents an exponential decay of the change rate, and the
third and the fourth terms indicate advective and diffusive migration of elevation perturbation, respectively. The coefficients of the latter two terms determine the relative strength between advection and diffusion, with the ratio defined as the Péclet number, $P_e$:

$$P_e = \frac{C_0 - D_0'}{D_0} \ell, \tag{14}$$

where $\ell$ is the length of a perturbation. If $P_e$ is much higher than 0, forward advection will dominate, and any perturbation of
ice thickness will only propagate downstream. This prohibits destabilization in the upper stream if thinning or glacier retreat initiates near the terminus. On the other hand, if $P_e \sim 0$ or is negative, either diffusion or backward advection takes place, and a thickness perturbation at the terminus can propagate upstream, changing the dynamics of the entire glacier. Hence, we consider a glacier with low $P_e$ more vulnerable than one with high $P_e$.

Combining Eqs. 1, 2, 7, and 8 with Eq. 14, we can express $P_e$ in terms of ice speed, elevation, and surface slope (see Text
S4 of Zheng et al., 2019, Eqs. 11 to 16 for derivation details):





$$P_e = \left[ \frac{(m+1)\alpha_0}{mH_0} - \frac{U_0'}{U_0} - \frac{H_0'}{H_0} + \frac{\alpha_0'}{\alpha_0} \right] \ell \qquad (15)$$

The final assumption we adopt in the model is $\frac{\partial \alpha_0}{\partial x} = \alpha_0' \approx 0$ since the estimated value from the data we use in this paper is essentially small. For example, $\alpha_0'$ of the glacier profile shown in Fig. 3 is only around $10^{-6}$–$10^{-7}$ m$^{-1}$ for the first 100 km. In practice, this assumption might be necessary since the last term in Eq. 15 is highly sensitive to local slope change and may not

reflect the overall mechanism a glacier follows to dissipate the perturbation. With this assumption, Eq. 15 can be reduced as:

$$\frac{P_e}{\ell} \approx \left[ \frac{(m+1)\alpha_0}{mH_0} - \frac{U_0'}{U_0} - \frac{H_0'}{H_0} \right]. \qquad (16)$$

Note we now express Péclet number as the form of $\frac{P_e}{\ell}$ since we do not focus on a particular perturbation length and instead plan to evaluate the general tendency for the ice flow to dissipate any length of perturbations. Compared to the past models, the expression of $P_e$ in this model has an additional term $\frac{U_0'}{U_0}$, implying its relationship to spatially changing basal conditions. This

extra dependency on glacier speed also suggests that $P_e$ is a changing variable and needs to be re-calculated if ice flow speeds up or slows down (see Discussion for more details).

With the same assumption about $\alpha_0'$, the expression of $J_0$ (Eq. 10) can be also reduced to:

$$J_0 \approx C_0 H_0' = (m+1)U_0 H_0', \qquad (17)$$

which is proportional to the product of ice speed and the gradient of ice thickness along the flowline. A typical glacier thins

toward the terminus, corresponding to a negative $H_0'$ and $J_0$. According to Eq. 13, a negative $J_0$ suggests that when a lubricating scenario takes place, the glacier will speed up to accommodate the change. From Eq. 9 we can see that the glacier will also get thickened (except at the divide) since thicker ice is sliding and replaces thinner ice downstream.

To summarize, two parameters $P_e$ and $J_0$ are derived from the 1-D basal lubrication model. $J_0$ represents the strength of initial response to basal lubrication, and $P_e$ gives insights into the mode of mass transport after elevation change occurs.

Glaciers with a high $|J_0|$ and a low $P_e$ ($\sim 0$ or negative) are more vulnerable to basal lubrication since reduced friction can lead to a high initial acceleration and elevation change rate, which will then propagate to the entire glacier via diffusion or negative advection.

## 3 Data and Methods for Validating the Model

To test if the model is suitable for evaluating marine-terminating glaciers, we derive observed $P_e/\ell$ and $J_0$ for outlet glaciers in

the GrIS and Austfonna Ice Cap, Svalbard. These two regions are selected primarily because surface elevation, bed elevation, and glacier speed data necessary for our calculation are publicly available. We compare the results with the NASA MEaSUREs ITS_LIVE glacier velocity records (Gardner et al., 2018, 2019) spanning over 20 years and determine if both $P_e/\ell$ and $J_0$ are indicative of the vulnerability to basal lubrication.





## 3.1 Greenland Ice Sheet

We use the data set published with Felikson et al. (2021), which provides well-constrained flowline data for Greenland's 141 marine-terminated glaciers and their branches (187 basins in total, Fig. 1). These glaciers scatter around the ice sheet and provide a diverse sampling over various climate and oceanic factors. The data set contains six primary flowline shapes for each glacier basin, with vertices sampled every 50 m along the flowlines. We use surface elevations at each vertex, sampled from the Greenland Ice Mapping Project (GIMP, Howat et al., 2014). The GIMP surface elevations come from multiple remote

sensing sources and are coregistered with elevations acquired by the Ice, Cloud, and land Elevation Satellite (ICESat), thus best representing the ice sheet elevations during 2003–2009. The flowline vertices also contain the glacier bed elevations, sampled from the BedMachine v3 subglacial topography (Morlighem et al., 2017). While BedMachine v3 uses the source data collected from 1993 to 2016, we assume that the bed elevations are stable over time and can represent any year in that period. To acquire $U_0$ and glacier speed change at each flowline vertex, we manually sample the annually-mosaicked ITS_LIVE glacier speed

data from 1998 and 2018, respectively. The ITS_LIVE data are derived from Landsat 4, 5, 7, and 8 images using the autoRIFT feature tracking software Gardner et al. (2018); Lei et al. (2021). Finally, each vertex of a flowline has the following key parameters: surface elevation, bed elevation, glacier speed in 1998, and speed difference between 1998 and 2018.

We prepare and process the input data for each flowline using the following steps:

1. Since the 1998 speed data do not cover the entire ice sheet, we remove flowlines with only 20 speed readings or less
from the input list.

2. Locate vertices with NoData speed values along the flowlines and perform a linear interpolation to fill the missing values.

3. Remove flowlines with only 280 valid vertices or less from the input list. A valid vertex should contain all key parameters and no NoData Values.

4. To avoid the effect of small sloping change, we smooth the surface elevation, bed elevation, glacier speed data, and
their derivatives using the Savitzky–Golay filter with a window size of 251 vertices (12.5 km) (Savitzky and Golay, 1964; Felikson et al., 2021). We do not apply the smoothing filter to data 0–3 km from the terminus due to insufficient sampling points within the window size. These unfiltered data will not be used for the next step.

5. Derive $P_e/\ell$ and $J_0$ along each flowline using Eqs. 17, 16, and parameters representative of the glacier geometry/speed from 1998. As we empirically derive $P_e/\ell$ and $J_0$ for each basin and compare them on the same plot, the results will be
insensitive to the selected value of $m$ and the sliding law (Felikson et al., 2021).

6. Compare the results with the speed change between 1998 and 2018.

## 3.2 Austfonna Ice Cap, Svalbard

We perform the same analysis for the marine-terminating glaciers of Austfonna, a polythermal Ice Cap located in NE Svalbard. Austfonna is only about 100 km wide and is considered to have a more uniform climate and oceanic factors than the GrIS, but



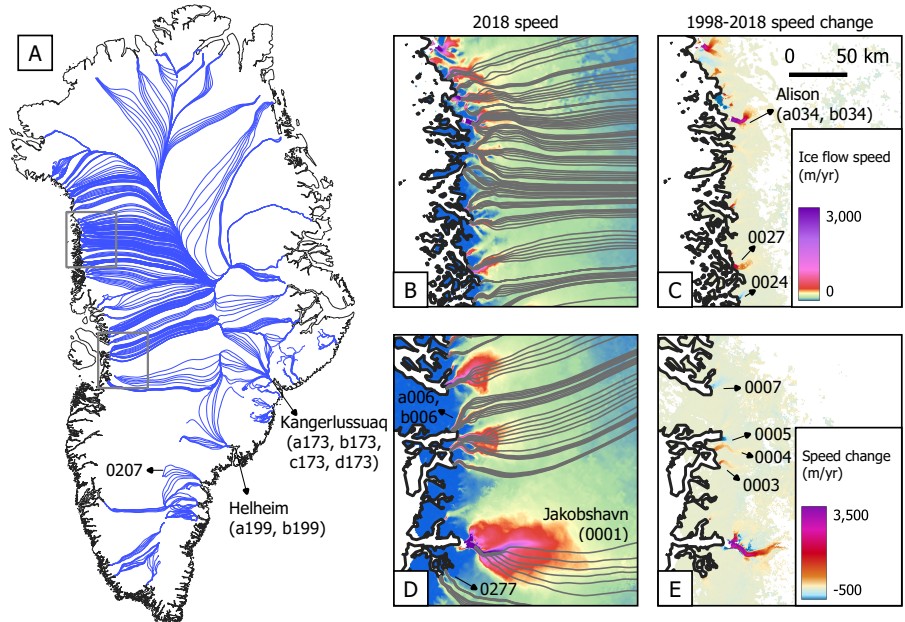

**Figure 1.** Greenland glacier flowlines used in this study. (A) Location of all selected flowlines from 104 outlet glaciers or glacier branches. Gray boxes indicate map locations for panels on the right. (B–C) The closer view of the flowline distribution, speed from 2018, and speed change during 1998 and 2018 at NW Greenland, a place with the most flowlines across the ice sheet. (D–E) Same panels as B–C but for W Greenland where Jakobshavn Isbræ is located in the bottom. The glacier speeds from 1998 and 2018 are both sampled from the ITS_LIVE data set. Major glaciers and most glaciers in the zoom-in panels are labeled with names and IDs used in the source data set.

its marine outlet glaciers have exhibited diverse speed histories in the past 20 years (Fig. 2). For instance, the glacier speed of Basin-3 (Storisstraumen) increased 45-fold during the past two decades, likely triggered by feedback between summer melt, crevasse formation, and basal lubrication (McMillan et al., 2014; Dunse et al., 2015; Gong et al., 2018). The other surge-type glaciers include Basin-1 (Bråsvellbreen) and Basin-17 (Etonbreen), and the last surge periods of both glaciers are around 1938 (Schytt, 1969; Hagen et al., 1993; Dowdeswell et al., 2008). The other glaciers of Austfonna do not have a surge history, but

many of them (e.g., Basin-2, -5, -7, and -10) have also significantly increased the flow speed since 1996, as seen from Fig. 2.

To calculate $P_e/\ell$ and $J_0$, we use the Ice Thickness Models Intercomparison eXperiment (ITMIX) data set (Farinotti et al., 2017), hosted by the International Association of Cryospheric Sciences (IACS). We use the Austfonna DEM from 1996 (Moholdt and Kääb, 2012), velocity from 1995–1996 (Dowdeswell et al., 2008), and ice thickness from 1996 (Dowdeswell et al., 1986; Farinotti et al., 2017), all included in the ITMIX data set. 8 out of 11 marine-terminated glaciers (Basin-1, -3, -4, -5, -6,

-7, -10, and -17; Fig. 2) are selected for the analysis; the exception are Basin-2, -8, and -9 due to their small length roughly equal to the smoothing window. We construct six glacier flowlines based on the 2018 glacier velocity from the ITS_LIVE annual velocity mosaics (Fig. 2B) and sample ITMIX glacier elevation, ice thickness, and glacier speed data every 50 m along





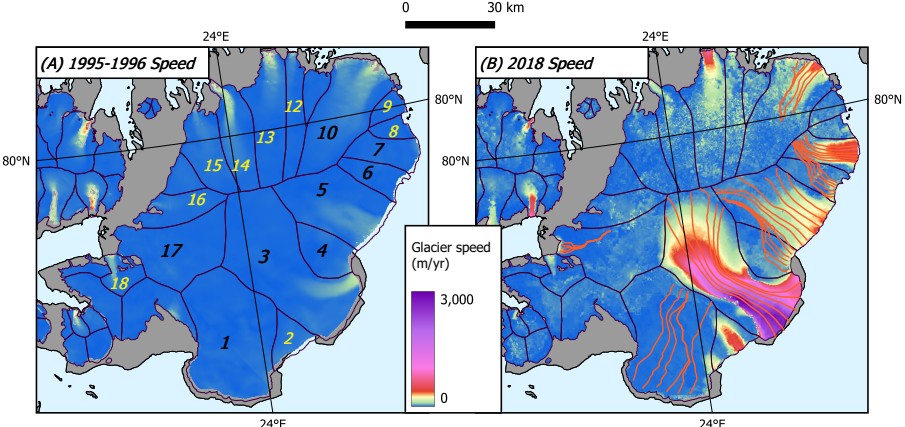

**Figure 2.** Austfonna Ice Cap, Svalbard, and glacier flowlines. (A) ITMIX glacier speed in 1995–1996. Each glacier basin is labeled with a number as per Dowdeswell et al. (2008), and glaciers with a black numbers are analyzed in this study. (B) ITS_LIVE mosaicked glacier speed in 2018. For each selected basin, we generate six flowlines and plot them on the map as red lines. Glacier outlines are from the Randolph Glacier Inventory (RGI) version 6.0 (RGI Consortium, 2017).

each flowline. Then we follow the same workflow for the outlet glaciers in GrIS (see the previous section) and finally compare $P_e/\ell$ & $J_0$ with the ITS_LIVE 2018 glacier speeds.

## 4 Results

### 4.1 Variation within a single basin

Figures 3 and 4 provide example results within a single basin, showing both input data and the values of $P_e/\ell$ and $J_0$ along six major flowlines. The average frontal speed at Glacier 0207 (65.17°N, 41.16°W, Figs. 1 & 3) has changed from ∼6000 m yr$^{-1}$ to ∼9500 m yr$^{-1}$ during the studied period, suggesting a destabilized status. The value of $P_e/\ell$ of individual flowlines ranges

from $8 \times 10^{-5}$ to $-2 \times 10^{-5}$ m$^{-1}$ for the first 20 km from the terminus, but the average value is more constrained roughly at 4–5 $\times 10^{-5}$ m$^{-1}$ for the first 10 km. Compared to $P_e/\ell$, $J_0$ changes more quickly throughout the first 20 km, from ∼600 m yr$^{-1}$ to -200 m yr$^{-1}$. The pattern of the 1998–2018 speed change resembles that of $J_0$ calculated using the 1998 data, indicating $J_0$ as a good predictor of glacier speedup. If we plot $J_0$ versus $P_e/\ell$ (Fig. 3F) along the first 10 km from the terminus, the average values will roughly form a line going vertically on the plot.

On the other hand, Glacier 0277 (Alangordliup Sermia, 68.95°N, 50.22°W, ∼30 km south of Jakobshavn Isbræ; Figs. 1 & 4) is more stable than Glacier 0207 since the amount of the speed change in the past two decades is only 0–40 m yr$^{-1}$ and is constrained at the first 6 km from the terminus. $P_e/\ell$ ranges from $2 \times 10^{-4}$ to $6 \times 10^{-4}$ m$^{-1}$ within the first 10 km, which is roughly 10 times the values from Glacier 0207. Also, the slow glacier speed in 1998 directly results in low $|J_0|$ (only ∼-10

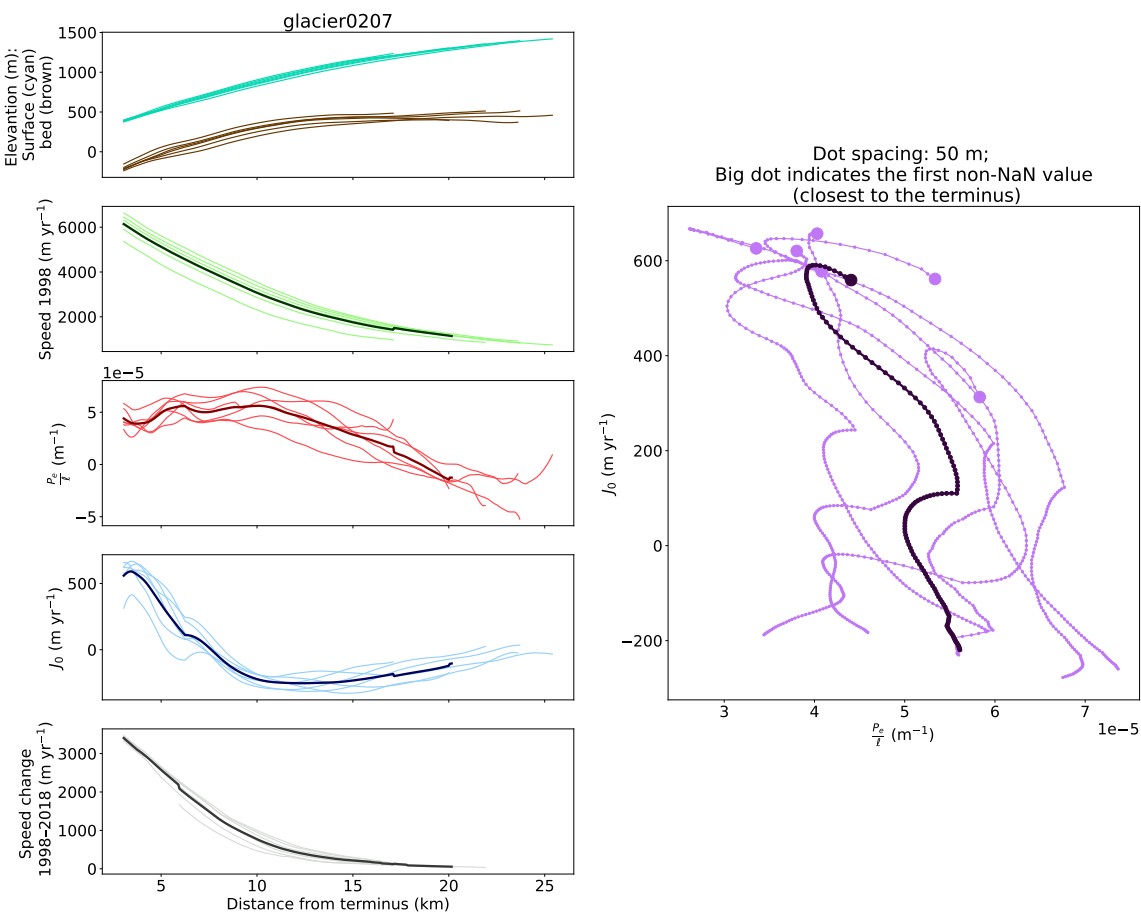

**Figure 3.** Example results from glacier 0207 (65.17°N, 41.16°W). (A) surface elevation (Cyan) and bed elevation (Brown). (B) surface speed in 1998. (C) $P_e/\ell$. (D) $J_0$. (E) Speed change between 1998 and 2018. These plots show all six flowlines profiles from a single basin in respect to the distance from the glacier terminus. The thick lines represent the average of all six flowlines. (F) $J_0$ versus $P_e/\ell$ along the first 10 km from the terminus. The big dots represent values at 3 km or the valid values closet to the terminus, and the small dots are plotted every 50 m along the flowline.

m yr$^{-1}$) compared to Glacier 0207. These results are supportive for Glacier 0277 having a stable condition during the study period. Interestingly, the speed change pattern now resembles $P_e/\ell$ instead of $J_0$ at the first 10 km. The glacier might have dealt with frontal or basal perturbation through advection, as indicated by a large $P_e$.

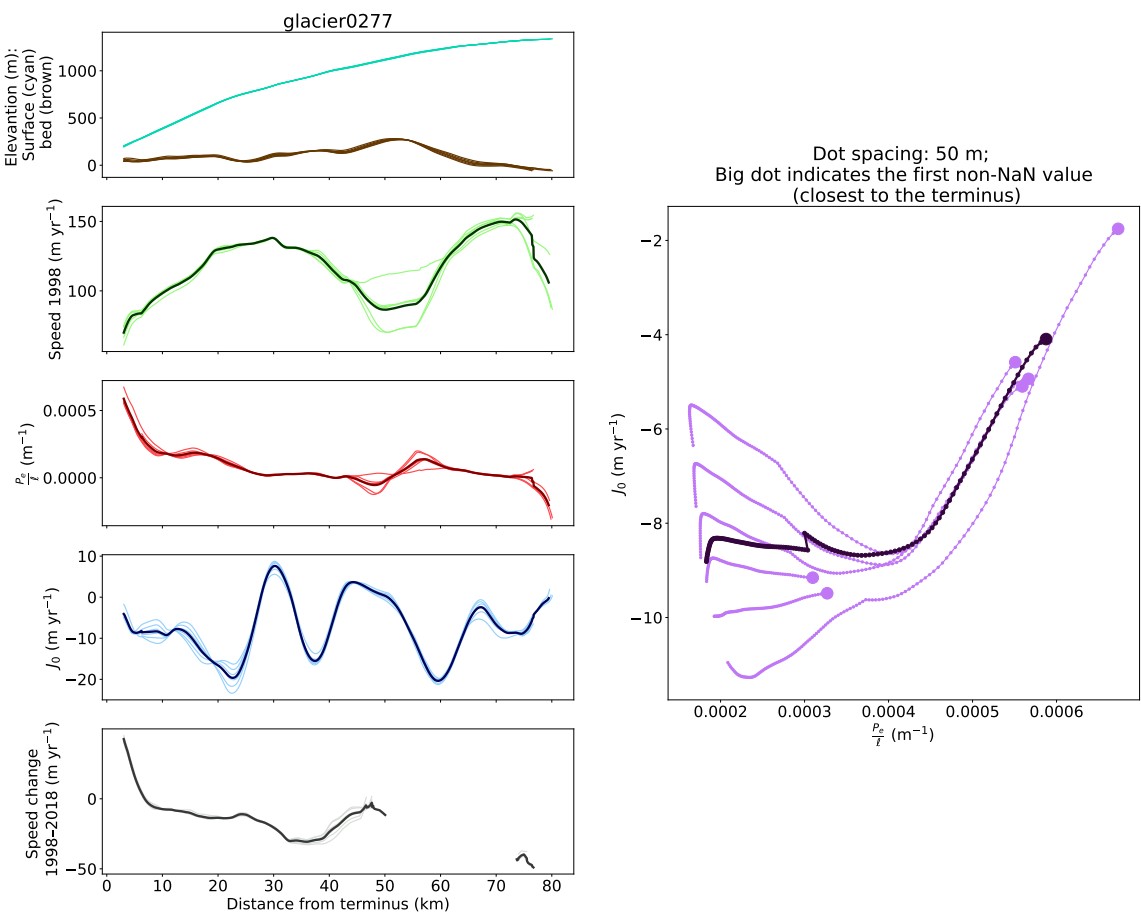

**Figure 4.** Example results from glacier 0277 (Alangordliup Sermia, 68.95°N, 50.22°W). See Fig. 3 for detailed description about each panel.

The results of the other GrIS and Austfonna glaciers are available in the Github-Zenodo supplemental materials (https://doi.org/10.5281/zenodo.5641953).

## 4.2 $P_e$ & $J_0$ versus glacier speed change

Due to the incomplete spatial coverage of ITS_LIVE data from 1998, only 104 out of 187 GrIS glacier basins have valid values of $P_e/\ell$ and $J_0$ at 3 km from the terminus (Fig. 1), and we select them for the following intercomparison. While most of these glaciers have sped up during the 20 years, 26 glaciers slowed down by up to -522 m yr⁻¹ (Fig. 5A). On the $J_0$–$P_e/\ell$ plot, each





ball-head-pin-like curve represents the average $J_0$ & $P_e/\ell$ values of one glacier basin at 3–5 km, color-coded based on their speed change at 3 km, and its head marks the values at 3 km as well (Fig. 5B). Glaciers with low speed change (pale color curves) tend to cluster around the area where $J_0 \approx 0$ m yr⁻¹ and $P_e/\ell > 0.001$ m⁻¹. Most of these curves are near horizontal on the plot, indicating a small change of $J_0$ and a large change of $P_e/\ell$ at the glacier front. Likewise, glaciers with high speed change (warm- or cold-color curves, including accelerated and decelerated glaciers) seem to cluster together on a different area where $J_0$ is much more negative and $P_e/\ell \approx 0$ m⁻¹. These curves generally show a vertical orientation indicating changing $J_0$ and constant low $P_e/\ell$ along the glacier flowline.

To further illustrate the clustering trend, we arbitrarily select a speed change threshold of $\pm 300$ m yr⁻¹ and classify the glaciers based on their speed change at 3 km. The value is determined in order to give each classification roughly the same number of samples. Note that glaciers with significant slowdown or speedup are classified into the same group because a glacier vulnerable to basal lubrication would also be sensitive to recovering basal conditions from a lubrication event. 54 glaciers have an absolute value of speed change $\geq 300$ m yr⁻¹, and the other 50 glaciers are considered more stable with an absolute value of speed change $< 300$ m yr⁻¹. We plot $J_0$ against $P_e/\ell$ using the values from 3 km as well as the Gaussian kernel density estimates of each classification for both $J_0$ and $P_e/\ell$ (Fig. 6). The results indicate that two classifications have a slightly different distribution for $J_0$ and $P_e/\ell$. The unstable glaciers (red group on Fig. 6) have $J_0$ and $P_e/\ell$ distributions peaked at $\sim -200$ m yr⁻¹ and $\sim 0.00003$ m⁻¹ respectively, while the the peaks of stable glaciers (blue group on Fig. 6) shift to higher values to $\sim -50$ m yr⁻¹ for $J_0$ and $\sim 0.00013$ m⁻¹ for $P_e/\ell$.

We adopt the same method from Fig. 5 to plot the results from 8 marine-terminating glaciers in Austfonna (Fig. 7). Since all glaciers have accelerated at the terminus for the past two decades, we adjust the color code so that blue represents low change and other colors represent high change. For Basin-3 and -5, there are no valid measurements at 3 km, and we only mark the first valid measurements from 7.3 and 6.7 km, respectively, as single points on the plot. Similar to GrIS, glaciers with higher speed change (Basin-3, -5, -7, and -10) roughly occupy the lower left side of the panel where $P_e/\ell$ and $J_0$ are small or more negative, and glaciers with lower speed change (Basin-1, -4, -6, and -17) fall on a different corner where $P_e/\ell$ and $J_0$ are larger. The only important difference is the scale of $|J_0|$ as all 8 glaciers have values between 0 and 10 m yr⁻¹, much less than that from GrIS (cf. Fig. 5, with $|J_0|$ ranging from 0 to over 1500 m yr⁻¹). This is because all 8 glaciers are slowly moving in 1996, resulting in a low $|J_0|$ according to Eq. 17. In addition, Basin-1 and -5 have similar $J_0$–$P_e/\ell$, but Basin-5 has a speed change roughly ten times more than Basin-1. Since Basin-1 (Bråsvellbreen) has a surge record back in 1936–1938 (Schytt, 1969) and is currently in the quiescent stage, its low $J_0$ & $P_e/\ell$ values might indicate a future instability when a surge is triggered internally or externally.

## 5 Discussion

### 5.1 Separation of glacier groups on the $J_0$–$P_e/\ell$ plot

The $J_0$–$P_e/\ell$ plot (Figs. 5–7) seems to capture the characteristics of glaciers vulnerable to basal lubrication. GrIS and Austfonna glaciers with more negative $J_0$ and $P_e/\ell$ in 1996–1998 are more likely to speed up in the next 20 years. However, this



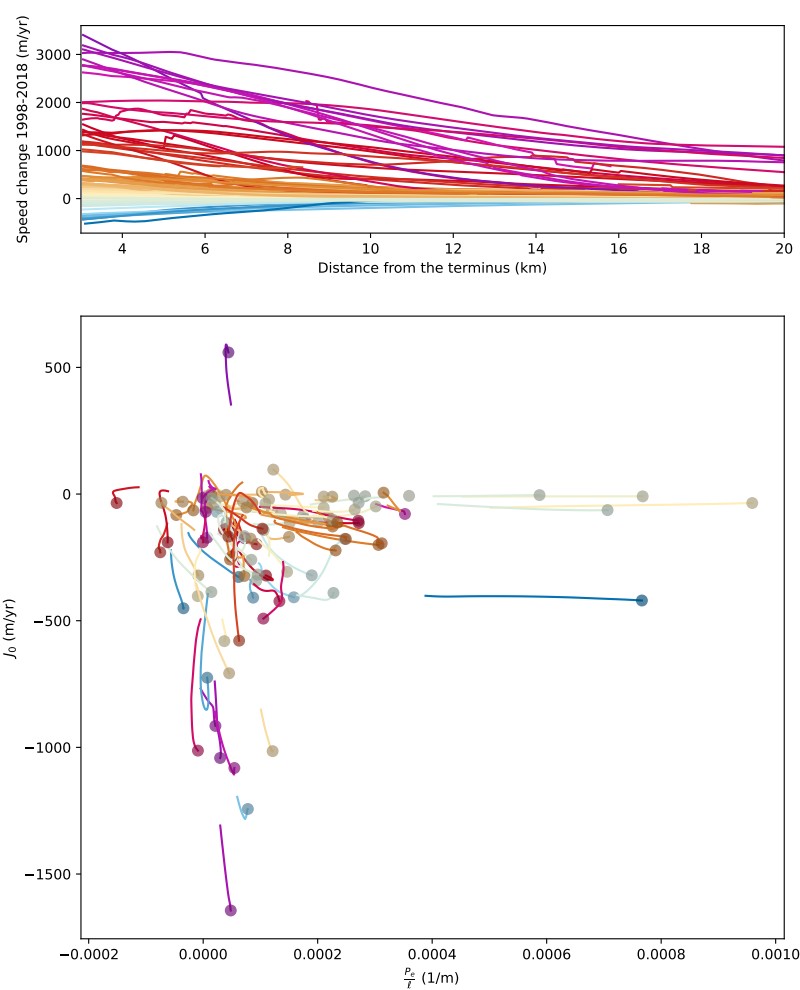

**Figure 5.** (A) Speed change along Greenland glacier flowlines within the first 20 km from the terminus. Each line represents the average value of a single glacier basin and is color coded based on the speed change value at 3 km (the first valid data point after the Savitzky–Golay filter is applied). (B) The average $J_0$ versus $P_e/\ell$ at 3–5 km from the terminus. The value at 3 km is marked with a big dot. Each line uses the same color code from (A).

trend is ambiguous for glaciers with $J_0 = -500$–$0$ m yr$^{-1}$ and $P_e/\ell = 0$–$0.0001$ m$^{-1}$ as some of them remain stable over 20 years while the others accelerated significantly (Figs. 5B and 6). This can be possibly attributed to two different reasons. First,





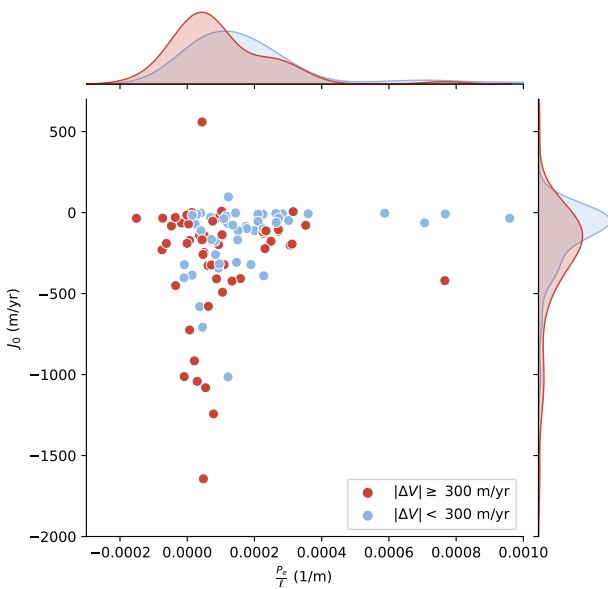

**Figure 6.** Distribution of $J_0$ and $P_e/\ell$ for Greenland's 104 glacier basins. This plot is similar to Fig. 5B but only shows the values at 3 km from the terminus. Each mark is classified based on the 300 m/yr threshold of speed change. For each class, the joint plots show the Gaussian kernel density estimate along both axes.

a low $P_0$ indicates a diffusion-dominating dynamics for dissipating elevation change, but a small $|J_0|$ prohibits much elevation change under a lubrication scenario (Eq. 9). As a result, the glacier flow would be insensitive to a lubrication event and accom-

modate any subsequent perturbations slowly. The second reason focuses on whether the basal lubrication takes place within the study period. Although melt-induced speedups are common for GrIS glaciers (e.g. van de Wal et al., 2008; Bartholomew et al., 2010; Kehrl et al., 2017; Rathmann et al., 2017; Seddik et al., 2019), not all 104 glacier basins on Fig. 6 have been studied well enough to identify when and where glacier responds to changing basal conditions. These two reasons suggest that the glaciers with low $|J_0|$ and $P_e/\ell$ are still likely susceptible to changing basal conditions but need more time to adjust glacier speed and

fully accommodate the lubrication.

The three surge-type glaciers of Austfonna (Basin-1, -3, and -17) do not cluster on Fig. 7. Compared to the other two glaciers, Basin-3 has a higher flow speed in 1995–1996, resulting in a slightly more negative $J_0$. It has an unusual long surge evolution over two decades as well: the ice flow speed gradually increased since the mid-1990s (Dowdeswell et al., 2008; McMillan et al., 2014) and reached a peak velocity of $\sim 6500$ m yr$^{-1}$ in 2013 (Dunse et al., 2015). The sustaining high flow speed has

been attributed to meltwater routing through crevasses formed during the speedup (Dunse et al., 2015; Gong et al., 2018). Since the additional support of surface melt can alter the behavior of a surge-type glacier by reaching a steady state balancing mass and enthalpy conservation through thinner and faster-moving ice (Benn et al., 2019), Basin-3 may have entered an ice stream-like regime with higher sensitivity to changing basal conditions. This might explain why Basin-3 is away from the other





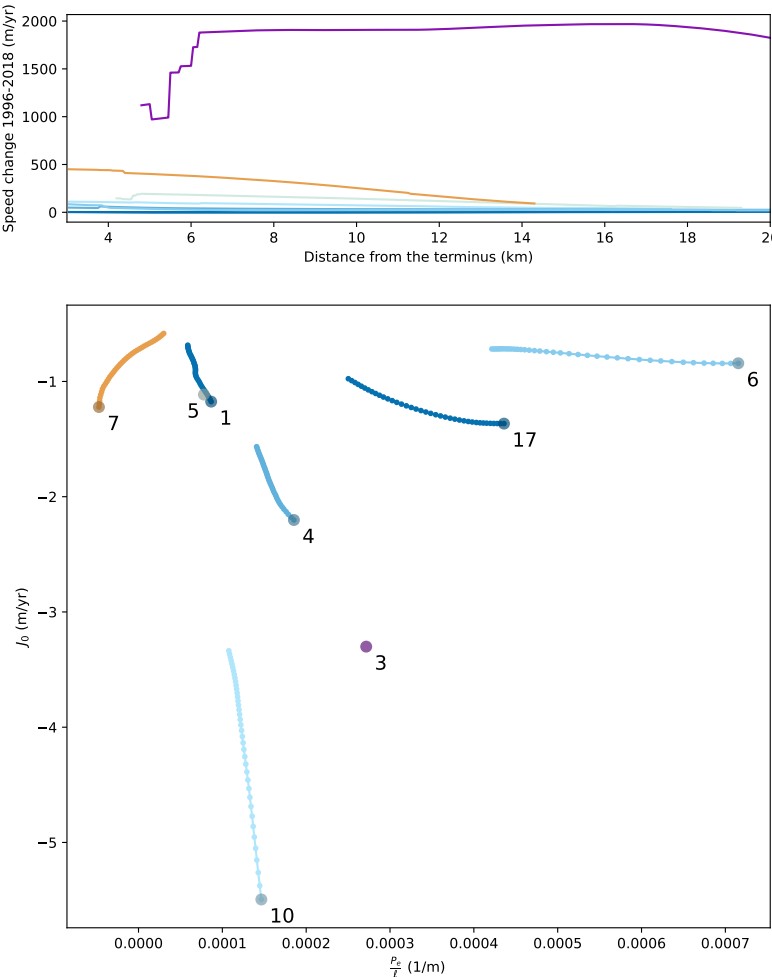

**Figure 7.** (A) Speed change along Austfonna glacier flowlines within the first 20 km from the terminus. Each line represents the average value of a single glacier basin and is color coded based on the speed change value at 3 km. (B) The average $J_0$ versus $P_e/\ell$ at 3–5 km from the terminus. The value at 3 km is marked with a big dot. Each line uses the same color code from (A) and is labeled by the glacier No. (see Fig. 2). Small dots are plotted every 50 m along each flowline. Note that for Basin-3 and -5, there is no valid measurement at 3 km, and only the first valid measurements (at 7.3 and 6.7 km respectively) are plotted.





two surge-type but currently quiescent glaciers on Fig. 7. Nevertheless, additional analysis and tests will be required before

inferring a general vulnerability for surge-type glaciers to basal lubrication.

## 5.2  Characteristics of glaciers vulnerable to basal lubrication

Both $P_e$ and $J_0$ depend on the ice thickness and flow speed (Eqs. 16 & 17), but with a different relationship. Assuming a monotonous decrease of glacier thickness toward the terminus (i.e., no overdeepening area), glaciers with thicker ice and a faster flow yield lower $P_e$ and higher $|J_0|$ and thus are more susceptible to basal lubrication. However, the thickness change

along the flowline ($H'$) has a competing contribution to $P_e$ and $J_0$: a greater change (i.e. a more negative $H'$) increases both $P_e$ and $|J_0|$. In this paper, all the GrIS and Austfonna glaciers with such a thickness change are slowly moving in 1996 or 1998, still yielding a low $|J_0|$ (Figs. 3 and 5). These glaciers may not likely be activated through intense diffusion (which is also suggested true for a terminus perturbation scenario in Felikson et al., 2021), but a collapse-like destabilization is still possible at the terminus or a localized region along the glacier. For a glacier with an overdeepening zone, increased ice thickness again

lowers $P_e$ and raises $|J_0|$, making the glacier more vulnerable to basal lubrication at the overdeepened area.

Despite having an additional associating factor $J_0$ in the model, inferences made to the Péclet number in this study is similar to the previous models based on the perturbation theory. A low or negative $P_e$ allows an ocean-induced terminus perturbation to propagate to a certain inland distance where $P_e$ becomes larger (Felikson et al., 2017, 2021). For an outlet glacier in the GrIS, it is probably common to have terminal perturbation and changing basal conditions in effect at the same time (as indicated by

Jakobshavn Isbræ for example; Joughin et al., 2008; Khazendar et al., 2019; Riel et al., 2021). In this case, $P_e$ reflects a general vulnerability to elevation perturbations and is indistinguishable from the source forcing. On the other hand, $J_0$ as a new term in the lubrication-induced perturbation model seems to be exclusively related to the basal sensitivity. Nevertheless, $J_0$ might still be a key factor for a glacier only subject to the ocean-ice interaction. As warm subsurface water-induced thinning debuttresses the glacier front and increases the longitudinal stretching and glacier speed (Holland et al., 2008), new crevasses can provide

additional routes for surface melt accessing to and lubricating the bed (Gagliardini and Werder, 2018; Gong et al., 2018). The investigation for surface strain rates indicates that these new crevasses can propagate to up to 1600 m high, corresponding to ~50 km away from the terminus for GrIS outlet glaciers (Poinar et al., 2015). Thus, $J_0$ can be used to evaluate the latter mechanism's impact, specifically for subsequent ice flow acceleration or the feedback to additional thinning. This oceanic forcing-induced basal lubrication seems to be important for marine-terminating glaciers to switch to and maintain a fast flow

over the years since land-terminating glaciers in the GrIS are thought to be insensitive to extensive meltwater forcing (Tedstone et al., 2013; Williams et al., 2020).

Our model does not consider the melt production from strain heating or geothermal heating at the bed. If included, induced glacier speedup due to basal lubrication can generate energy to melt basal ice (Strozzi et al., 2017b), further increasing $K_1$ and leading to a higher glacier speed and thinning rate than what Eqs. 6 and 12 indicate.





### 5.3 Feedback from basal lubrication

One implication from the lubrication-induced perturbation model is that both $P_e$ and $J_0$ are temporally changing variables. As glacier speed increases due to basal lubrication, $P_e$ will be closer to zero, and $|J_0|$ will become larger, resulting in status more sensitive to any following change of basal conditions. A potential example to illustrate this feedback is Vavilov Ice Cap, Severnaya Zemlya, Russia. The western marine outlet of this ice cap was moving at less than 1 km yr⁻¹ with no apparent summer speedups just before a surge-like collapse took place (Willis et al., 2018). The collapse initiated when the terminus advanced into weak marine sediments, bringing the glacier speed to a maximum of ∼9 km yr⁻¹ in summer 2015, and then the ice flow started to slow down but with a significant seasonal variation (Willis et al., 2018; Zheng et al., 2019). During the collapse, Péclet number at the thinning center reduced by at least 40%, suggesting a shift to ice stream-like dynamic regime more susceptible to basal lubrication (Zheng et al., 2019). However, this transition is not necessarily irreversible since the bed would also be sensitive to a refreezing or efficient draining event, increasing $P_e$ and decreasing $|J_0|$ back to the pre-perturbation level. Such a cycle is probably happening on a yearly basis as many GrIS glaciers have seasonal speedups closely bonded to summer melt (e.g. Palmer et al., 2011; Sundal et al., 2013; Rathmann et al., 2017). Still, For a multi-year destabilization like Vavilov, it is uncertain whether such a shutdown can completely revert the glacier dynamics since the ice thickness, another critical parameter controlling $P_e$ and $J_0$ in our model, has changed much during the collapse as well.

As noted in the case of Vavilov, dynamic thinning caused by the lubricated bed can also consequently change the dynamic regime, create another feedback loop. Thinning would decrease the ice thickness and increase the surface slope, potentially raising $P_e$ and $|J_0|$. Unlike the acceleration feedback from the previous paragraph, this feedback circle seems to be milder as a glacier would gradually switch to advection and prevent further inland thinning. However, dynamic thinning may also contribute to glacier retreat (Thomas and Bentley, 1978; Wood et al., 2021) and the subsequent debuttressing and speedup events. The net effect for dynamic thinning to glacier vulnerability to basal conditions remains ambiguous based on this view and suggests a future research topic since dynamic discharge in GrIS will likely continue to contribute significant ice loss in the near future (Mouginot et al., 2019; Choi et al., 2021).

## 6  Conclusions

Based on the new lubrication-induced 1-D perturbation model, we show that a lubricated bed can initiate a thinning perturbation and destabilize the entire glacier if a particular combination of glacier thickness, thickness gradient, and flow speed is met. The model identifies two controlling physical quantities $P_e/\ell$ (Péclet number divided by the characteristic length) and $J_0$ (essentially the product of glacier speed and thickness gradient). We use observational data from 1996–1998 and derive these numbers for 104 and 8 marine-terminating glaciers in Greenland Ice Sheet and Austfonna Ice Cap, Svalbard, respectively. The results show that $P_e/\ell$ and $J_0$ correlate to the flow speed change during 1996/1998 and 2018, matching the model prediction. Glaciers with thick ice and a fast flow result in low $P_e$ and negative $J_0$, and reduced basal friction leads to initial speedup and thinning, which can propagate further inland via diffusion. For glaciers in the Greenland Ice Sheet subject to ocean-ice interactions, this new model indicates multiple feedback cycles that make glaciers more sensitive to changing basal conditions.
Finally, this study highlights glaciers classified as vulnerable to lubricated beds (low $P_e$ and high $|J_0|$) but with no significant speed change during the past two decades. Frequent monitoring is suggested for these glaciers because they might be more

325  prone to future instabilities and affect the projected sea level rise.

*Code and data availability.*  All the data, workflows, documentation, supplemental figures, plotting scripts, and Python code for this study are available on the Github repository "`whyjz/pejzero`" (https://github.com/whyjz/pejzero). The pejzero release version 0.1 is archived on Zenodo: https://doi.org/10.5281/zenodo.5641954, where a detailed description about additional assets (large files that Github cannot track) is available. The pejzero repository has been rendered as Jupyter Book pages at https://whyjz.github.io/pejzero/ and is Binder-ready for full

330  reproducibility. The Greenland glacier flowline data and scripts prepared by Felikson et al. (2021) are available at https://doi.org/10.5281/zenodo.4284759 and http://doi.org/10.5281/zenodo.4284715. Specific instructions on data retrieval and ingestion (including the flowline, ITMIX, and ITS_LIVE data) can be found on the `Fig*.ipynb` files in the pejzero repository.

*Author contributions.*  Whyjay Zheng conceptualizes the study, analyzes the data, and writes the manuscript.

*Competing interests.*  The author declares that they have no conflict of interest.

335  *Acknowledgements.*  This study is partially supported by the Jupyter meets the Earth, an NSF EarthCube funded project (grant No. 1928406 and 1928374). Comments from Dr. Matthew Pritchard and Dr. Michael Willis greatly improved the quality of the paper. The author thanks Dr. Denis Felikson for addressing questions about the use of the GrIS flowline data. The author also acknowledges the National Snow and Ice Data Center QGreenland package (Moon et al., 2021) as an extremely convenient tool for inter-comparing the GrIS data sets and preparing maps presented in this paper.




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
