# Peer review of "Glacier geometry and flow speed determine how Arctic marine-terminating glaciers respond to lubricated beds"

_The Cryosphere, 2021_

## Referee Comment (RC1)

**Review of the article entitled "Glacier geometry and flow speed determine how Arctic marine-terminating glaciers respond to lubricated beds"**

**1    General comments**

This manuscript presents further development on the precedingly applied perturbation theory. Rather than using a perturbation in elevation, the author is introducing a basal friction term which will be the source of the perturbation. The development of the model leads to the definition of two key parameters, $J_0$ represent the strength of the initial response to basal lubrication and $P_e$ gives insight on the longer term mode of mass transport. The author used multi-sourced data from 1996-1998 from glaciers in Greenland and Austfonna ice cap to compute $P_0$ and $J_e$ and compare those to the observe velocity changes in the following twenty years. The application of this 1D model allows to highlight glaciers for which a combination of thickness and initial velocity will render said glacier vulnerable to a lubrication of its bed. The study indicates that for some ocean terminating glaciers in the Greenland ice sheet, multiple feedbacks could make glaciers more sensitive to changing basal condition.

The introduction of basal friction in the perturbation theory model allows to introduce a more realistic way to perturb the model and observe its evolution. This improvement allows to draw conclusions on the stability of glaciers to a lubrication event with respect to its $P_0$ and $J_e$ which ultimately are based on its geometry and velocity before the perturbation. The result unfortunately show that there is quite a large area of the $J_0$ against $\frac{P_e}{l}$ graph in which the behaviour of the glaciers is uncertain and their classification as vulnerable or not to basal lubrication is not clear. The author propose two hypothesis that would explain this clustering on the graph with quite different effect in term of acceleration. Finally, the author presents some characteristics which would render a glacier vulnerable to lubrication and the potential feedback due to said lubrication.

The description of the model, and development of its equations is clear and well described but some of the figures could be clarified for better readability. As a colourblind reader I have a hard time with the colour choice of Figures 5 and 7. This is not vital to the understanding of the paper as the difference between pale and other colours is still

readable but it will help to control the colourscale and/or add a colourbar to help with readability. On panel (B) of these figures, it might also be better to use a smaller marker than the dot to allow a better readability of the lines themselves.

There are a few more points that might need clarification in the paper as those were not completely clear to me.

- In the description of the data treatment, it is stated that the velocities are interpolated on NoData points and afterwards that a valid vertex should not contain any NoData Values, I expect that at this point the interpolated velocities are not considered as NoData anymore, is that right?

- It is clear particularly on the Austfonna example that the flowlines are displaced due to the changes in velocity, I wonder how these changes affect the model result and why the flowlines were computed from the 2018 dataset and not from the ITMIX one.

- Only marine terminating glacier are investigated here, is there any limitation of the model that prevents to study land terminating glaciers, or was it a choice of the author?

- Ultimately, most of the study investigates $J_0$ as close to the front as possible, it would be interested on the reasons behind that choice in the paper.

- On figure 6, I noted that the spread of the Gaussian kernels is different for both classes of glacier, more spread on $J_0$ and more defined peak on $\frac{P_e}{l}$ for glaciers with high acceleration and the other way around for the more stable glaciers. Can that be explained by the model? If yes it would probably be worth discussing that pattern.

- Line 285, it is stated that GrIS land terminating glaciers are insensitive to meltwater forcing. However there are some modelling study that seem to disagree with that statement [e.g. Gagliardini and Werder, 2018]

**2 Specific comments**

Bellow is a list of more specific comments throughout the manuscript given with line numbers:

- Line 1: "flow" seems misplaced here, isn't "discharge of ice" sufficient?

- Line 11: I do not completely understand the usage of "forms" here.

- Line 16: I am not sure of the meaning of "where a good portion" is it for most of Greenland glaciers, or most of the marine terminating glaciers?

- Line 23: Should "subsurface ocean water" be specified here?

- Line 28: Sentence on this line is unclear and could be rephrased.

- Table 1: $K_1$ here is a function of $x$ and $t$ but it is stated in the text that it is a constant, perhaps the notation in the table should reflect that.

- Table 1: The primes (') are not defined here perhaps it would be worth defining them as $x$ derivative here.

- Line 86: I suspect that $fracdhdt$ stands for $fracdH_1dt$

- Figure 3: In this figure and the following, the panel lettering is missing in the figure.

- Line 207: I would prefer the "$J_0$ against $\frac{P_e}{l}$" notation than the one used here and further down (line 233, 239).

- Line 210: It seems that a zero is missing in the value of $P_e/l$.

- Figure 5: In this figure and following the units should be changed to be consistent with the text.

- Line 292: I am not sure of the meaning of status here.

- Line 306: Isn't "and" missing, "and create an other feedback..."

**References**

Olivier Gagliardini and M. A. Werder. Influence of increasing surface melt over decadal timescales on land-terminating greenland-type outlet glaciers. *J. Glaciol.*, 64(247):1–11, aug 2018. doi: 10.1017/jog.2018.59.

---

## Referee Comment (RC2)

*Glacier geometry and flow speed determine how Arctic marine-terminating glaciers respond to lubricated beds* by Whyjay Zheng, 2022.

**Overview**
This study investigates the interannual impact of increased basal lubrication on glacier flow using a 1-D physical framework and tested on >100 glacier basins in Greenland and Austfonna Ice Cap Svalbard. Within the model framework, they determine that both the Péclet number over length (*Pe/l*) and a metric proportional to the product of speed and ice thickness gradient, termed $J_o$. $J_o$ reflects the initial response to basal lubrication, and *Pe/l* reflects a general vulnerability to an elevation perturbation. The model results predict that glaciers are most sensitive to increased basal lubrication (that is, they will undergo greater acceleration given perturbed basal conditions) when $J_o$ is relatively high and *Pe/l* is minimized or negative. Finally, these two quantities are calculated from observational data in 1996/1998 and compared to the acceleration observed along flowlines by comparing earlier speeds to those observed in 2018 from ITS_LIVE. They conclude that given a certain combination of glacier thickness, thickness gradient, and speeds are met, enhanced basal lubrication can destabilize and accelerate the full length of the glacier. This is an interesting approach aimed at identifying glaciers that are vulnerable to destabilization and provides useful information on how the baseline glacier geometry informs potential basal vulnerabilities. The manuscript is well-written and presents a creative approach to constraining a complex science question. The figures complementary to the text, and I appreciate the effort to document and archive model code and results through Github and interactive Jupyter notebooks. With some expanded motivation, and polished analysis and figures, this paper could make a valuable contribution to the cryosphere/glaciology community. Below, I've first listed my main comments/concerns, followed by minor comments.

**Main Comments**
1.) The premise of this work is centered on the concept of a potential *permanent* change to glacier basal conditions and constraining how the related effects on glacier dynamics (thinning and acceleration). It would be helpful to introduce the physical basis for such a change, rather than surge-type glaciers, including explicitly describing what such conditions would look like in reality. I understand that the spatially uniform increase in basal lubrication (reduction in basal friction, or K term) is not meant to imitate reality but is useful as a modeling tool. However, given the strong seasonality observed at Greenland glaciers in response to summertime meltwater and evolving subglacial conditions, what kind of environment meets the criteria of a "permanent change"? One with greater seasonal oscillations between efficient and inefficient drainage systems, one with continuous drainage and elevated basal water pressures throughout the year, or another scenario entirely? There seems to be a missing connection here that makes it somewhat challenging to contextualize how the findings of the paper inform our understanding of future climatic conditions on ice sheets/ice caps.

2.) The conclusions include some statements that extent beyond the results presented in the manuscript. do not seem entirely supported by the findings in the manuscript. For example, the phrase in the conclusion on line 239 states that: "The $J_o$–*Pe*/ℓ plot (Figs. 5–7) seems to capture the characteristics of glaciers vulnerable to basal lubrication. GrIS and Austonna glaciers with

more negative $J_o$ and $Pe/\ell$ in 1996–1998 are more likely to speed up in the next 20 years." This argument can be made for the GIS glaciers based on the distributions shown in Figure 6, but it is far from obvious for Austfonna glaciers show in Figure 6. I think, with the limited sample of glaciers and subset that include surge types, there is not enough information to assert a distinction based on $J_o$ and $Pe/l$ alone. The conclusion should reflect this uncertainty. Even for the n=104 glaciers in Greenland, where distributions show a tendency for greater accelerations at basins with low/more negative $J_o$ and $Pe/l$, the text should be careful to emphasize that this reflects results at a specific distance alone a glacier flow line and may not be representative of the entire glacier length.

3.) As addressed in the text, terminus retreat is also a common source of acceleration, especially at Greenland glaciers, and retreat impacts are indistinguishable from increased basal lubrication within the presented framework. I think it would be highly valuable to include net retreat when considering acceleration over the 1998-2018 period. For example, how does speed increases observed within subsets with low $Pe/l$ /negative $J_o$ and minimal retreat compare to acceleration observed at glaciers with low $Pe/l$ /negative $J_o$ but significant retreat? Showing that these variables are still applicable to acceleration in the absence of terminus retreat would strengthen the significance of the study.

It also may be worthwhile to evaluate the two groups of glaciers (here divided based on acceleration greater than or less than 300 m/yr) based on the percent increase in speed (such as > or <= 10%), rather than an absolute (300 m/yr) threshold.

**Secondary/Minor Comments**

Figures
-All axis labels and unit font sizes need to be enlarged.

-Please include lettered labels (a, b, c, etc.) on the subplots corresponding to the labels mentioned in the figure captions.

-Include a scale bar for zoomed inserts in Figure 1 and in Figure 2.

-Please also include legends for your figures. This includes a color bar for speed increases in Figure 5 and 7.

*Figure 3*
Køge Bugt (glacier 0207 in Figure 3) has retreat around 2 km between 1998 and 2018. This site also appears to have the greatest $J_o$ values of the Greenland sample (shown in Figure 5), which would imply the most diminished sensitivity to respond to basal lubrication. This seems at odds with the statement on line 193, that states that Jo is a good predictor of glacier speed up at this basin.

*Figure 6*
Are the differences between the two groups' distributions statistically significant?

*On the 3km flowline position analyses*
Why is this position (3 km for 1998-2018 speed change and mean 3-5 km parameters) used for the majority of the analyses? Can you provide justification for why this distance from the terminus is most representative of glacier sensitivity to basal lubrication?

*Line 232*
The range in Jo should be to -1500 m/yr, not 1500, correct?

---

## Author Comment (AC1)

**Author response for Reviewer 1**

I would like to thank the reviewer for commenting on the manuscript constructively with valuable ideas and insights. I appreciate the time and effort you made that will surely improve the quality of the paper. Below are my detailed responses to the Reviewer Comments. The original text is in **black** and my response is in **green**.
* * *
**1 General comments**

This manuscript presents further development on the precedingly applied perturbation theory. Rather than using a perturbation in elevation, the author is introducing a basal friction term which will be the source of the perturbation. The development of the model leads to the denition of two key parameters, $J_0$ represent the strength of the initial response to basal lubrication and $P_e$ gives insight on the longer term mode of mass transport. The author used multi-sourced data from 1996-1998 from glaciers in Greenland and Austfonna ice cap to compute $P_0$ and $J_e$ and compare those to the observe velocity changes in the following twenty years. The application of this 1D model allows to highlight glaciers for which a combination of thickness and initial velocity will render said glacier vulnerable to a lubrication of its bed. The study indicates that for some ocean terminating glaciers in the Greenland ice sheet, multiple feedbacks could make glaciers more sensitive to changing basal condition.

The introduction of basal friction in the perturbation theory model allows to introduce a more realistic way to perturb the model and observe its evolution. This improvement allows to draw conclusions on the stability of glaciers to a lubrication event with respect to its $P_0$ and $J_e$ which ultimately are based on its geometry and velocity before the perturbation. The result unfortunately show that there is quite a large area of the $J_0$ against $P_e/\ell$ graph in which the behaviour of the glaciers is uncertain and their classification as vulnerable or not to basal lubrication is not clear. The author propose two hypothesis that would explain this clustering on the graph with quite dierent effect in term of acceleration. Finally, the author presents some characteristics which would render a glacier vulnerable to lubrication and the potential feedback due to said lubrication.

The description of the model, and development of its equations is clear and well described but some of the figures could be clarified for better readability. As a colourblind reader I have a hard time with the colour choice of Figures 5 and 7. This is not vital to the understanding of the paper as the difference between pale and other colours is still readable but it will help to control the colourscale and/or add a colourbar to help with readability. On panel (B) of these figures, it might also be better to use a smaller marker than the dot to allow a better readability of the lines themselves.

Thank you for the suggestion. I was aware of the colorblind readability for most of the plots but obviously did not tune them to the best. I have redesigned the colormap used in Figures 5

and 7. Now the colormap is based on Roma with a few tweaks about the transparency and the red zone so it can nicely show an asymmetrical pattern of the speed change. I have also added a colormap scale in the lower panel and changed the marker size as suggested. I hope this improves the figure, but please let me know if there is still something that could be done for readability. I'd appreciate it very much.

There are a few more points that might need clarification in the paper as those were not completely clear to me.
• In the description of the data treatment, it is stated that the velocities are interpolated on NoData points and afterwards that a valid vertex should not contain any NoData Values, I expect that at this point the interpolated velocities are not considered as NoData anymore, is that right?
The NoData vertices that cannot be interpolated will be kept, so there is a possibility that the interpolated velocity still contains NoData at the end of the flowline. I have added the following sentence in the data treatment for improved clarity: "We do not extrapolate the glacier speed; therefore, the NoData vertices at both ends of the flowline are still preserved after this step."

• It is clear particularly on the Austfonna example that the flowlines are displaced due to the changes in velocity, I wonder how these changes affect the model result and why the flowlines were computed from the 2018 dataset and not from the ITMIX one.
The ITMIX data set contains only flow speed but not flow velocity (i.e. only scalar values available), making it not ideal for constructing glacier flowline. Nevertheless, one of the ITS_LIVE-derived Basin-3 flowlines (ID #36, the northernmost flowline at Basin-3 in Figure 2B) matches the ITMIX flow pattern. Thus, the model results should reflect the average of the changing flow patterns.

• Only marine terminating glacier are investigated here, is there any limitation of the model that prevents to study land terminating glaciers, or was it a choice of the author?
Yes, it is a choice of the author as land-terminating glaciers should follow the same physical framework (speedups triggered by basal lubrication and diffusion thinning) as marine-terminating glaciers do. The main reason for this choice is that GrIS's marine-terminating glaciers have shown significant mass loss through dynamic thinning, while land-terminating glaciers mainly drain the mass by a negative surface mass balance (see the first paragraph of the main text and the references therein). As a result, marine-terminating glaciers seem to have a larger impact on their mass budget under basal lubrication than land-terminating glaciers do, so they become ideal targets for testing the physical framework presented in this study.

• Ultimately, most of the study investigates $J_0$ as close to the front as possible, it would be interested on the reasons behind that choice in the paper.
There are two reasons for this choice:
1. The crevasse/mouline formation is more likely to occur at the terminus region than at the divide region (see the updated text in the manuscript for references), which means the terminus region is more likely to be lubricated by meltwater routing.

2. $J_0$ is proportional to the initial flow speed (Eq. 17). The ice flow away from the front is usually small, and glacier basins tend to have a similar small value of $J_0$ if using observations from the upper stream, making the metric less useful to determine vulnerability.

I have added a paragraph in Section 4.2 about the justifications above.

• On figure 6, I noted that the spread of the Gaussian kernels is different for both classes of glacier, more spread on $J_0$ and more defined peak on $P_e/\ell$ for glaciers with high acceleration and the other way around for the more stable glaciers. Can that be explained by the model? If yes it would probably be worth discussing that pattern.

I have included the frontal retreat data and updated the analysis associated with Figure 6 (see my response to Reviewer 2 for details). Now it looks that the observation described in the comment is linked to (1) various intensity of terminus retreat for each glacier (i.e. some glaciers accelerated due to retreat, not basal lubrication); and (2) small sample size causing a wide spread as seen from the Gaussian kernel (see the attached figure in the response to Reviewer 2). I have rewritten a great portion of Section 4.2 to reflect these changes and thoughts.

• Line 285, it is stated that GrIS land terminating glaciers are insensitive to meltwater forcing. However there are some modelling study that seem to disagree with that statement [e.g. Gagliardini and Werder, 2018]

I agree with this observation and have reviewed the corresponding paragraph. Since this paragraph mainly talks about marine-terminating glaciers, I have removed the latter half of the sentence about land-terminating glaciers for a more focused discussion. The Gagliardini and Werder 2018 paper has been cited elsewhere in the original manuscript.

**2 Specific comments**

Bellow is a list of more specific comments throughout the manuscript given with line numbers:

• Line 1: "flow" seems misplaced here, isn't "discharge of ice" sufficient?

Yes, "discharge of ice" is sufficient. Changed.

• Line 11: I do not completely understand the usage of "forms" here.

This sentence has now changed to "A combined factor of ice thickness, surface slope, and initial flow speed physically determines how much and how fast glaciers respond to lubricated beds in terms of speed, elevation, and terminus change.

• Line 16: I am not sure of the meaning of "where a good portion" is it for most of Greenland glaciers, or most of the marine terminating glaciers?

Thank you for pointing out this ambiguity. I removed "where a good portion of…" and rewrote this sentence as: "At the Greenland Ice Sheet (GrIS), dynamic discharge of marine-terminating glaciers accounts for 66% of the region's total mass loss (Mouginot et al., 2019)."

• Line 23: Should "subsurface ocean water" be specified here?

Changed as suggested for improved clarity.

• Line 28: Sentence on this line is unclear and could be rephrased.
To improve clarity, this part of the text has been modified to: "Outside of the GrIS, the primary drivers of the dynamic ice loss remain largely uncertain, although significant melt-induced lubrication and speedup events have been identified around the Arctic."

• Table 1: $K_1$ here is a function of x and t but it is stated in the text that it is a constant, perhaps the notation in the table should reflect that.
Changed the notation in the table from $K_1(x, t)$ to $K_1$ with a note that $K_1$ is assumed constant.

• Table 1: The primes (') are not defined here perhaps it would be worth defining them as x derivative here.
Added a sentence in the Table 1 caption for clarity.

• Line 86: I suspect that *fracdhdt* stands for *fracdH₁dt*
Yes, that's correct. Changed to $dH_1/dt$.

• Figure 3: In this figure and the following, the panel lettering is missing in the figure.
Added the missing panel letters in Figures 3, 4, 5, and 7.

• Line 207: I would prefer the "$J_0$ against $P_e/\ell$" notation than the one used here and further down (line 233, 239).
Changed the notation to "$J_0$ versus $P_e/\ell$ plot".

• Line 210: It seems that a zero is missing in the value of $P_e/\ell$.
Thank you for pointing this typo out. Changed to $P_e/\ell > 0.0001$.

• Figure 5: In this figure and following the units should be changed to be consistent with the text.
Changed all velocity-like units (e.g., glacier speed change and $J_0$) from m/yr to m yr$^{-1}$ and Pe/$\ell$ units from 1/m to m$^{-1}$, including those label in Figures 1,2, 5-7.

• Line 292: I am not sure of the meaning of status here.
Changed "resulting in status" to "making the glacier."

• Line 306: Isn't "and" missing, "and create an other feedback.."
Corrected.

**References**

Olivier Gagliardini and M. A. Werder. Influence of increasing surface melt over decadal timescales on land-terminating greenland-type outlet glaciers. J. Glaciol., 64(247):111, aug 2018. doi: 10.1017/jog.2018.59.

---

## Author Comment (AC2)

**Author response for Reviewer 2**

I would like to thank the reviewer for commenting on the manuscript constructively with valuable ideas and insights. I appreciate the time and effort you made that will surely improve the quality of the paper. Below are my detailed responses to the Reviewer Comments. The original text is in **black** and my response is in **green**.
* * *
**Overview**

This study investigates the interannual impact of increased basal lubrication on glacier flow using a 1-D physical framework and tested on >100 glacier basins in Greenland and Austfonna Ice Cap Svalbard. Within the model framework, they determine that both the Péclet number over length ($P_e/\ell$) and a metric proportional to the product of speed and ice thickness gradient, termed $J_0$. $J_0$ reflects the initial response to basal lubrication, and $P_e/\ell$ reflects a general vulnerability to an elevation perturbation. The model results predict that glaciers are most sensitive to increased basal lubrication (that is, they will undergo greater acceleration given perturbed basal conditions) when $J_0$ is relatively high and $P_e/\ell$ is minimized or negative. Finally, these two quantities are calculated from observational data in 1996/1998 and compared to the acceleration observed along flowlines by comparing earlier speeds to those observed in 2018 from ITS_LIVE. They conclude that given a certain combination of glacier thickness, thickness gradient, and speeds are met, enhanced basal lubrication can destabilize and accelerate the full length of the glacier. This is an interesting approach aimed at identifying glaciers that are vulnerable to destabilization and provides useful information on how the baseline glacier geometry informs potential basal vulnerabilities. The manuscript is well-written and presents a creative approach to constraining a complex science question. The figures complementary to the text, and I appreciate the effort to document and archive model code and results through Github and interactive Jupyter notebooks. With some expanded motivation, and polished analysis and figures, this paper could make a valuable contribution to the cryosphere/glaciology community. Below, I've first listed my main comments/concerns, followed by minor comments.

**Main Comments**

1.) The premise of this work is centered on the concept of a potential *permanent* change to glacier basal conditions and constraining how the related effects on glacier dynamics (thinning and acceleration). It would be helpful to introduce the physical basis for such a change, rather than surge-type glaciers, including explicitly describing what such conditions would look like in reality. I understand that the spatially uniform increase in basal lubrication (reduction in basal friction, or K term) is not meant to imitate reality but is useful as a modeling tool. However, given the strong seasonality observed at Greenland glaciers in response to summertime meltwater and evolving subglacial conditions, what kind of environment meets the criteria of a "permanent change"? One with greater seasonal oscillations between efficient and inefficient

drainage systems, one with continuous drainage and elevated basal water pressures throughout the year, or another scenario entirely? There seems to be a missing connection here that makes it somewhat challenging to contextualize how the findings of the paper inform our understanding of future climatic conditions on ice sheets/ice caps.

The original motivation to look into a permanent change of basal conditions comes from the observations of several significant glacier speedups. As in the introduction: "(These events change the basal conditions) via creating a highly crevassed glacier surface which makes meltwater reach the bed more easily." These events are mostly located in the European/Russian Arctics, but a similar finding that excessive melt expedites moulin/crevasse formation has also been proposed for Greenland Ice Cap (Hoffman et al., 2018, https://doi.org/10.1002/2017GL075659). Regardless of whether a Greenland glacier has seasonally varying subglacial conditions, the basal friction can be subject to an interannual (and perhaps continuous) decrease due to the formation of these extra melt routes. As stated in the manuscript, the interannual impact of basal lubrication is less studied than the seasonal signal of speed variation. With this in mind, this paper tries to develop a simple framework to understand the varying response to these interannual dynamic changes. I have updated the manuscript (mostly for the introduction section) with the thoughts above to hopefully help contextualize the findings of the paper.

2.) The conclusions include some statements that extent beyond the results presented in the manuscript. do not seem entirely supported by the findings in the manuscript. For example, the phrase in the conclusion on line 239 states that: "The $J_o$–Pe/$\ell$ plot (Figs. 5–7) seems to capture the characteristics of glaciers vulnerable to basal lubrication. GrIS and Austonna glaciers with more negative $J_o$ and Pe/$\ell$ in 1996–1998 are more likely to speed up in the next 20 years." This argument can be made for the GIS glaciers based on the distributions shown in Figure 6, but it is far from obvious for Austfonna glaciers show in Figure 6. I think, with the limited sample of glaciers and subset that include surge types, there is not enough information to assert a distinction based on $J_o$ and Pe/l alone. The conclusion should reflect this uncertainty. Even for the n=104 glaciers in Greenland, where distributions show a tendency for greater accelerations at basins with low/more negative $J_o$ and Pe/l, the text should be careful to emphasize that this reflects results at a specific distance alone a glacier flow line and may not be representative of the entire glacier length.

Agreed. I have modified the discussion section so that it aligns with the results better and more conservatively. The updated manuscript now:
- States that the $J_o$ versus Pe/$\ell$ plot characterizes GrIS glaciers in terms of their vulnerability to basal lubrication, but for surge-type glaciers in Austfonna we need more data to fully find out the relationship between $J_o$ / Pe and glacier accleration.
- Reflects the uncertainty of the GrIS results in terms of the specific terminal distance analyzed in the paper.

3.) As addressed in the text, terminus retreat is also a common source of acceleration, especially at Greenland glaciers, and retreat impacts are indistinguishable from increased basal lubrication within the presented framework. I think it would be highly valuable to include net retreat when considering acceleration over the 1998-2018 period. For example, how does

speed increases observed within subsets with low Pe/l /negative Jo and minimal retreat compare to acceleration observed at glaciers with low Pe/l /negative Jo but significant retreat? Showing that these variables are still applicable to acceleration in the absence of terminus retreat would strengthen the significance of the study.

Thank you for your insightful suggestion. I have retrieved the terminal retreat data from Wood et al. (2021, https://doi.org/10.1126/sciadv.aba7282, Data repository http://doi.org/10.7280/D1667W) and made a comparison with $J_0$ and $P_e/\ell$. This figure shows how $J_0$ and $P_e/\ell$ scatter for all glaciers with terminal retreat < 0.5 km:

[Figure]

The red group now only contains 7 glaciers since the other accelerating glaciers typically have a significant terminal retreat, but the rest of the data points clearly separate two glacier groups based on their $J_0$ and $P_e/\ell$. I have updated the manuscript with this additional analysis and results, including the data description, two extra figures, discussion text, and supplementary Jupyter Book pages.

It also may be worthwhile to evaluate the two groups of glaciers (here divided based on

acceleration greater than or less than 300 m/yr) based on the percent increase in speed (such as > or <= 10%), rather than an absolute (300 m/yr) threshold.

Using the percent increase instead of the absolute increase results in a more compacted histogram with a few glaciers exceeding +100% of the speed change. It is thus harder to justify the separation of two groups of glaciers regardless of the chosen threshold value. I have added the histogram to the Figure 6 Jupyter Book page for a comparison with the existing histogram made using the absolute increase.

**Secondary/Minor Comments**

Figures
-All axis labels and unit font sizes need to be enlarged.
Done.

-Please include lettered labels (a, b, c, etc.) on the subplots corresponding to the labels mentioned in the figure captions.
Added the missing panel letters in Figures 3, 4, 5, and 7.

-Include a scale bar for zoomed inserts in Figure 1 and in Figure 2.
Both figures already have a scale bar for the zoomed inserts, and instead, I added an extra sentence in both captions clarifying what panels use the scale bar. For Figure 2, I also added an extra panel C showing the geographical location of Austfonna Ice Cap.

-Please also include legends for your figures. This includes a color bar for speed increases in Figure 5 and 7.
Done.

Figure 3
Køge Bugt (glacier 0207 in Figure 3) has retreat around 2 km between 1998 and 2018. This site also appears to have the greatest Jo values of the Greenland sample (shown in Figure 5), which would imply the most diminished sensitivity to respond to basal lubrication. This seems at odds with the statement on line 193, that states that Jo is a good predictor of glacier speed up at this Basin.

As long as $J_0$ is away from zero, basal lubrication would lead to an initial forcing of elevation change and further perturb the dynamic discharge. Køge Bugt has a $|J_0|$ = ~500 m yr$^{-1}$ around the terminus, which seems to be sufficiently high to cause such a forcing. What makes this glacier interesting is the sign of $J_0$ since this is the only glacier outlet with $J_0$ > 100 m yr$^{-1}$ among 187 basins. A positive $J_0$ requires a decreasing ice thickness from terminus to upstream (Eq. 17), which is unusual for a typical Greenland outlet glacier. As Figure 3 is meant to provide a typical case of what a glacier with flow speed change looks like, I agree that Køge Bugt may not be an ideal example for that and have replaced it with Jakobshavn Isbræ (glacier 0001) for this figure. I have also edited all the corresponding descriptions and analyses in the manuscript.

Figure 6

Are the differences between the two groups' distributions statistically significant?
I have performed a two-sample, two-sided Kolmogorov–Smirnov test for both $J_0$ and Pe/$\ell$, and both the test statistics indicate two groups are from different distributions (p-value = 0.003 and 0.006, respectively). I have added the statements above and relevant analysis in Section 4.2 and the corresponding Jupyter Book supplemental pages.

On the 3km flowline position analyses
Why is this position (3 km for 1998-2018 speed change and mean 3-5 km parameters) used for the majority of the analyses? Can you provide justification for why this distance from the terminus is most representative of glacier sensitivity to basal lubrication?
You can find my explanation as to why it should be good to use the data as close to the terminus as possible in my response to Reviewer 1. Practically, the data closest to the terminus cannot be properly smoothed using the Savitzky–Golay filter (Section 3.1) due to an insufficient window length, and I arbitrarily discard the data between 0 and 3 km away from the terminus to avoid potential bias from that, making the 3 km parameters the closest among all data analyzed in this study. I use the 3 km parameters for the following quantitative analyses (e,g, Figure 6 and the significance test above). The 3-5 km parameters are solely used for illustration in Figures 3, 4, 5, and 7 which provide an idea about how these values change along the flowline. I have experimented with different length segments and determined that showing the first two km of the valid data helps recognize the parameter pattern and change direction the best without being confused by too many data lines crossing each other in the figure (especially for Figure 5).

Line 232
The range in Jo should be to -1500 m/yr, not 1500, correct?
It should be 1500 m/yr because the absolute value of $J_0$ is discussed here. For clarity, I have added a note in the paragraph below Eq. 13 where the $|J_0|$ notation first appears in the text.